# Addressing cluster-constant covariates in mixed effects models via likelihood-based boosting techniques

**Colin Griesbach** [1]*, **Andreas Groll**[2], **Elisabeth Bergherr**[1]

**1** Department of Medical Informatics, Biometry and Epidemiology Friedrich-Alexander-University Erlangen-Nürnberg, Erlangen, Germany, **2** Faculty of Statistics, TU Dortmund, Dortmund, Germany

* colin.griesbach@fau.de

## Abstract

Boosting techniques from the field of statistical learning have grown to be a popular tool for estimating and selecting predictor effects in various regression models and can roughly be separated in two general approaches, namely gradient boosting and likelihood-based boosting. An extensive framework has been proposed in order to fit generalized mixed models based on boosting, however for the case of cluster-constant covariates likelihood-based boosting approaches tend to mischoose variables in the selection step leading to wrong estimates. We propose an improved boosting algorithm for linear mixed models, where the random effects are properly weighted, disentangled from the fixed effects updating scheme and corrected for correlations with cluster-constant covariates in order to improve quality of estimates and in addition reduce the computational effort. The method outperforms current state-of-the-art approaches from boosting and maximum likelihood inference which is shown via simulations and various data examples.

## 1 Introduction

Linear mixed models [1] proved to be a very popular tool for analysing data with repeated measurements, especially clustered longitudinal data from clinical surveys. Nevertheless, they are applicable to much broader fields and various overviews can be found in [2–4]. Fitting these models can be achieved with a variety of R packages available [5, 6] and classical methods for inference like tests [7] or selection criteria [8, 9] have been developed.

In order to use mixed models for prediction analysis, various approaches to regularized regression like lasso [10, 11] and boosting techniques [12] have been proposed. Lasso type approaches can be found in [13] for linear and in [14] for generalized linear mixed models. Boosting in general can be distinguished between gradient boosting [15, 16] and likelihood-based boosting [17, 18]. Both boosting methods are capable of fitting mixed models and for the latter an extensive framework has been proposed towards this matter in [19–21] and is included in the R package GMMBoost [22] available on CRAN. Apart from improving prediction analysis, component-wise boosting methods are due to an iterative and component-wise fitting process suitable for high dimensional data and implicitly offer variable selection. Good

**Data Availability Statement:** All relevant data are within the manuscript and its Supporting information files.

**Funding:** E.B.: Deutsche Forschungsgemeinschaft (DFG), Projekt WA 4249/2-1 https://www.dfg.de/en/index.jsp Volkswagen Foundation Freigeist

Fellowship https://www.volkswagenstiftung.de/en
The funders had no role in study design, data
collection and analysis, decision to publish, or
preparation of the manuscript.

**Competing interests:** The authors have declared
that no competing interests exist.

insights into component-wise boosting can be found in [23] for gradient boosting and in [24] for gradient and likelihood-based boosting as well. Please note that when talking about boosting, we always refer to the component-wise variant.

However, the `bGLMM` algorithm from the `GMMBoost` package tends to struggle with cluster-constant covariates, e.g. baseline covariates like gender or treatment group in longitudinal studies. The specified selection and updating procedure of the `bGLMM` algorithm tends to favour cluster-varying covariates while the simultaneously updated random intercepts partly account for effects actually evolving from cluster-constant covariates. As shown in Fig 1, this malfunction already occurs in a very basic data example with the popular Orthodont dataset, which is available in various R packages. The dataset depicts the evolution of an orthodontal measurement of 27 children and contains two covariates. A basic linear mixed model with random intercepts returns the two coefficient estimates $\hat{\beta}^{\text{lme}}_{\text{gender}} = -2.32$ by `lmer` and $\hat{\beta}^{\text{b}}_{\text{gender}} = 0.00$ by `bGLMM` for the effect of the cluster-constant covariate gender. The reason for this difference becomes clear when looking at the random intercepts, where `bGLMM` tends to compensate the missing effect for gender by assigning every female subject a random intercept lowered by 2.32. Although the structure of the Orthodont data set is very simple and does not require boosting, it is evident that the described weak spot of `bGLMM` is not confined to more complex datasets and thus can occur for any clustered data containing cluster-constant covariates.

We therefore propose an updated algorithm with various changes in order to avoid the phenomenon of random intercepts growing too quickly. These changes include the usage of smaller starting values and weaker random-effects updates to prevent the random effects from growing too fast as well as undocking the random effects update from the fixed effects boosting scheme, which guarantees a fair comparison between the single covariates for the fixed effects. Most importantly, we introduce a correction step for the random effects estimation to avoid possible correlations with observed covariates. The contribution of the present work is therefore a novel and better performing boosting algorithm regarding both estimation accuracy and runtime for mixed models, particularly in the presence of cluster-constant covariates. The algorithm not only solves the prescribed identification issues but in addition states the only regularization approach for mixed models, which explicitly accounts for estimation bias arising from possible correlations between random and regularized effects. While existing approaches bypass these issues by excluding affected covariates from the regularization

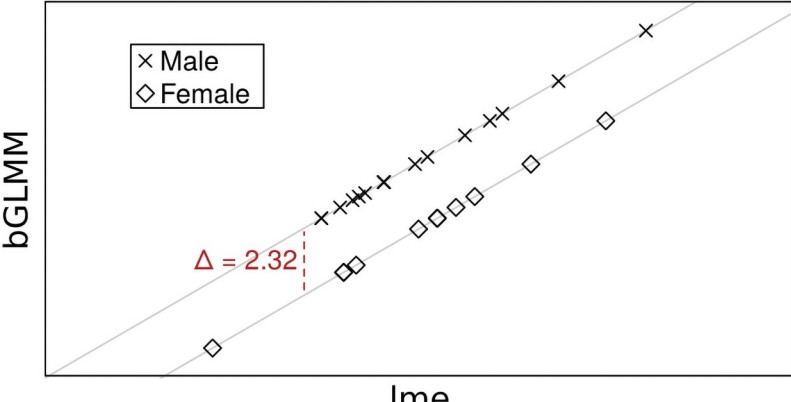

**Fig 1. Comparison between random intercept estimates by `lmer` and `bGLMM` for Orthodont.**

approach, the presented algorithm addresses the phenomena directly by correcting falsely estimated random effects.

The remainder of the paper is structured as follows: Section 2 formulates the underlying model and the updated boosting algorithm as well as a detailed discussion of the changes. The algorithm is then evaluated and compared to other regularization approaches using an extensive simulation study described in Section 3. As an illustrating data example we have chosen the Primary Biliary Cirrhosis data, which further underlines the strengths and weaknesses of the compared methods and is discussed in Section 4. Finally, the results and possible extensions are discussed.

## 2 Methods

We propose a novel and improved boosting algorithm for linear mixed models in the following subsections.

### 2.1 Model specification

For clusters $i = 1, \ldots, n$ with observations $j = 1, \ldots, n_i$ we consider the linear mixed model

$$y_{ij} = \beta_0 + \boldsymbol{x}_{ij}^T \boldsymbol{\beta} + \boldsymbol{z}_{ij}^T \gamma_i + \varepsilon_{ij},$$

with covariate vectors $\boldsymbol{x}_{ij}^T = (x_{ij1}, \ldots, x_{ijp})$ and $\boldsymbol{z}_{ij}^T = (z_{ij1}, \ldots, z_{ijq})$ referring to the fixed and random effects $\boldsymbol{\beta}$ and $\boldsymbol{\gamma}_i$, respectively. The random components are assumed to follow normal distributions, i.e. $\varepsilon_{ij} \sim \mathcal{N}(0, \sigma^2)$ for the model error and $\gamma_i \sim \mathcal{N}^{\otimes q}(\boldsymbol{0}, \boldsymbol{Q})$ for the random effects. This leads to a cluster-wise notation

$$\boldsymbol{y}_i = \beta_0 \boldsymbol{1} + \boldsymbol{X}_i \boldsymbol{\beta} + \boldsymbol{Z}_i \gamma_i + \varepsilon_i$$

with $\boldsymbol{y}_i = (y_{i1}, \ldots, y_{in_i})^T$, $\boldsymbol{1} = (1, \ldots, 1)$, $\boldsymbol{X}_i = (\boldsymbol{x}_{i1}, \ldots, \boldsymbol{x}_{in_i})^T$, $\boldsymbol{Z}_i = (\boldsymbol{z}_{i1}, \ldots, \boldsymbol{z}_{in_i})^T$ and $\varepsilon_i = (\varepsilon_{i1}, \ldots, \varepsilon_{in_i})$. Finally, we get the common matrix notation

$$\boldsymbol{y} = \beta_0 \boldsymbol{1} + \boldsymbol{X}\boldsymbol{\beta} + \boldsymbol{Z}\gamma + \varepsilon \tag{1}$$

of the full model with observations $\boldsymbol{y} = (\boldsymbol{y}_1^T, \ldots, \boldsymbol{y}_n^T)^T$, design matrices $\boldsymbol{X} = [\boldsymbol{X}_1^T, \ldots, \boldsymbol{X}_n^T]^T$ and the block-diagonal $\boldsymbol{Z} = \mathrm{diag}(\boldsymbol{Z}_1, \ldots, \boldsymbol{Z}_n)$. The random components $\varepsilon = (\varepsilon_1^T, \ldots, \varepsilon_n^T)^T$ and $\gamma = (\gamma_1^T, \ldots, \gamma_n^T)^T$ have corresponding covariance matrices $\sigma^2 \boldsymbol{I}_N$ and $\mathrm{diag}(\boldsymbol{Q}, \ldots, \boldsymbol{Q})$ where $\boldsymbol{I}_N$ is the $N = \Sigma n_i$ dimensional unit matrix.

In order to perform likelihood inference, let $\boldsymbol{\vartheta} = (\beta_0, \boldsymbol{\beta}^T, \gamma^T)$ denote the effects and $\boldsymbol{\phi} = (\sigma^2, \tau)$ information of the random components, where $\boldsymbol{\tau}$ contains the values of $\boldsymbol{Q}$. The marginal log-likelihood of the model can be obtained via

$$\ell(\boldsymbol{\vartheta}, \phi) = \sum_{i=1}^{n} \log \int \boldsymbol{F}(\boldsymbol{y}_i | \boldsymbol{\vartheta}, \phi) p(\gamma_i | \phi) d\gamma_i,$$

where $f(\cdot | \boldsymbol{\vartheta}, \phi)$ and $p(\cdot | \phi)$ denote the normal densities of the model error and the random effects. Laplace approximation following [25] results in the penalized log-likelihood

$$\ell^{\mathrm{pen}}(\boldsymbol{\vartheta}, \phi) = \sum_{i=1}^{n} \log \boldsymbol{F}(\boldsymbol{y}_i | \boldsymbol{\vartheta}, \phi) - \frac{1}{2} \sum_{i=1}^{n} \gamma_i^T \boldsymbol{Q}^{-1} \gamma_i, \tag{2}$$

which is going to be maximized simultaneously for $\boldsymbol{\vartheta}$ and $\phi$ by likelihood-based boosting-techniques discussed in the following subsection.

## 2.2 Boosting algorithm

The lbbLMM (likelihood-based boosting for linear mixed models) algorithm iteratively fits the linear mixed model (1) via component-wise likelihood-based boosting. The fitting procedure in general is carried out by Fisher-scoring [26], a variant of Newton's optimization method [27], which iteratively optimizes a given cost function based on quadratic approximations. It therefore obtains updates based on first order and second order derivatives, which are, in the context of Fisher scoring, represented by score vector and Fisher matrix of the underlying cost function. We first give a brief description of the algorithm and discuss the single steps in more detail in the following subsection.

   **Algorithm** lbbLMM

- **Initialize** estimates with starting values $\hat{\boldsymbol{\vartheta}}^{[0]}$ and $\hat{\phi}^{[0]}$. Choose total number of iterations $m_{\text{stop}}$ and step length $v$.

- **for** $m = 1$ to $m_{\text{stop}}$ **do**

- **step1: Update fixed effects**
  For $r = 1, \ldots, p$ define $\boldsymbol{\beta}_r := (\hat{\beta}_0^{[m-1]}, \hat{\beta}_r^{[m-1]})^T$ with $\hat{\beta}_r^{[m-1]}$ denoting the $r$th component of $\hat{\boldsymbol{\beta}}^{[m-1]}$. Compute score vector and Fisher matrix

$$s_r(\boldsymbol{\beta}_r) = \frac{\partial \ell^{\text{pen}}}{\partial \boldsymbol{\beta}_r}, \quad \boldsymbol{F}_r(\boldsymbol{\beta}_r) = -\mathbb{E}\left[\frac{\partial^2 \ell^{\text{pen}}}{\partial \boldsymbol{\beta}_r \partial \boldsymbol{\beta}_r^T}\right]$$

with respect to the current intercept $\hat{\beta}_0^{[m-1]}$ and the $r$th linear effect $\hat{\beta}_r^{[m-1]}$. Obtain $p$ possible updates

$$\boldsymbol{u}_r = \boldsymbol{F}_r(\boldsymbol{\beta}_r)^{-1} s_r(\boldsymbol{\beta}_r) \in \mathbb{R}^2$$

and find the best performing component $* \in \{1, \ldots, p\}$ minimizing AIC or BIC. This yields the update $\boldsymbol{u}_* = (u_0, u_*)$ containing the update $u_*$ for the effect $*$ with corresponding intercept update $u_0$. Receive $\hat{\beta}_0^{[m]}, \hat{\boldsymbol{\beta}}^{[m]}$ by updating

$$\hat{\beta}_0^{[m]} = \hat{\beta}_0^{[m-1]} + vu_0,$$

$$\hat{\beta}_r^{[m]} = \begin{cases} \hat{\beta}_r^{[m-1]} & \text{if } r \neq *, \\ \hat{\beta}_r^{[m-1]} + vu_* & \text{if } r = *, \end{cases} \quad r = 1, \cdots, p. \tag{3}$$

- **step2: Update random effects**
  Update random effects using an additional Fisher scoring step based on the penalized log-likelihood by calculating

$$s_{\text{ran}}(\gamma) = \frac{\partial \ell^{\text{pen}}}{\partial \gamma}, \quad \boldsymbol{F}_{\text{ran}}(\gamma) = -\mathbb{E}\left[\frac{\partial^2 \ell^{\text{pen}}}{\partial \gamma \partial \gamma^T}\right]$$

and weakly updating

$$\hat{\gamma}^{[m]} = \hat{\gamma}^{[m-1]} + v\boldsymbol{C}\boldsymbol{F}_{\text{ran}}(\gamma)^{-1} s_{\text{ran}}(\gamma).$$

The incorporation of the correction matrix $\boldsymbol{C}$ at this step is crucial and its derivation is discussed further below.

- **step3: Update variance-covariance-components**
  Update variance-covariance-components

$$\hat{Q}^{[m-1]} \to \hat{Q}^{[m]}$$

using an approximate EM-algorithm.

- **end for**

- **Stop** the algorithm at the best performing $m_*$ with respect to the specified information criterion. Return $\hat{\boldsymbol{\vartheta}}^{[m_*]}$ and $\hat{\phi}^{[m_*]}$ as the final estimates.

## 2.3 Computational details of the algorithm

We give a stepwise description of the computational details of the `lbbLMM` algorithm. For simplicity, we omit iteration indices and hats indicating estimated values whenever appropriate.

**2.3.1 Starting values.**   The parameters actually underlying the selection process are necessarily set to zero, thus $\hat{\boldsymbol{\beta}}^{[0]} = \mathbf{0}$. Initial intercept and model error are set to $\hat{\beta}_0^{[0]} = \bar{y}$, $\hat{\sigma}^{2[0]} = \mathrm{Var}(y)$ and random effects are initialized as $\hat{\gamma}^{[0]} = 0$ with small covariance-matrix, e.g. $\hat{Q}^{[0]} = \mathrm{diag}(0.1, \ldots, 0.1)$. An alternative approach which is also proposed in [19] would be fitting a standard linear mixed model for intercept and random effects

$$y = \beta_0 \mathbf{1} + Z\gamma + \varepsilon$$

by using e.g. the function `lmer` from the R package `lme4` and extracting the starting values from the model fit.

**2.3.2 Fixed effects boosting process.**   The computation of the $r$th update is straight forward by calculating

$$s_r(\boldsymbol{\beta}_r) = \sigma^{-2} \tilde{X}_r^T (y - \boldsymbol{\eta}), \quad F_r(\boldsymbol{\beta}_r) = \sigma^{-2} \tilde{X}_r^T \tilde{X}_r,$$

where $\tilde{X} = (1, X_{\bullet r})$ is a $N \times 2$ matrix containing a column of ones and the $r$th column of $X$ associated with the $r$th covariate and $\boldsymbol{\eta}$ denoting the current fit. This leads to $p$ possible parameter vectors $\boldsymbol{\vartheta}_r$, where only the intercept and $r$th component received an update according to $u_r$. The best performing component is the one leading to minimal $\mathrm{AIC}_r$ or $\mathrm{BIC}_r$ [28, 29] given by

$$\mathrm{AIC}_r \quad = -2 \sum_{i=1}^{n} \log F(y_i | \boldsymbol{\vartheta}_r, \phi) + 2\mathrm{df},$$

$$\mathrm{BIC}_r \quad = -2 \sum_{i=1}^{n} \log F(y_i | \boldsymbol{\vartheta}_r, \phi) + \log(n)\mathrm{df}.$$

Here, $\mathrm{df} = \phi + \#\{i \le p : \beta_i \neq 0\}$ denotes the model complexity according to the marginal likelihood where $\#\phi$ is the total number of variance-covariance parameters in $\phi$.

**2.3.3 Random effects update.**   By calculating

$$s_{\mathrm{ran}}(\gamma) = \frac{\partial \ell^{\mathrm{pen}}}{\partial \gamma}, \quad F_{\mathrm{ran}}(\gamma) = -\mathbb{E}\left[\frac{\partial^2 \ell^{\mathrm{pen}}}{\partial \gamma \partial \gamma^T}\right]$$

a weak and corrected update

$$\hat{\gamma}^{[m]} = \hat{\gamma}^{[m-1]} + v\boldsymbol{CF}_{\mathrm{ran}}(\gamma)^{-1} s_{\mathrm{ran}}(\gamma).$$

for the random effects is obtained. Note that this differs from the approach in [19] as the random effects are updated separately and in addition also receive an update scaled by the step length $v$. The weak update ensures that the random effects don't grow to quickly compared to the fixed effects. The disentanglement of the random effects update from the fixed effects updating scheme on the other hand guarantees a fair comparison of the single fixed effects, where the random effects do not play a crucial role. In addition, the Fisher matrix

$$\boldsymbol{F}_{\mathrm{ran}}(\gamma) = \mathrm{diag}(\boldsymbol{F}_1, \cdots, \boldsymbol{F}_n), \quad \boldsymbol{F}_i = \sigma^{-2}\boldsymbol{Z}_i^T\boldsymbol{Z}_i + \boldsymbol{Q}^{-1}$$

has block-diagonal form making the inversion much easier and thus strongly reducing the computational effort.

**2.3.4 Deriving the correction matrix $C$.** The single random intercepts or random slopes are corrected independently of each other using distinct sets of covariates. For the correction of the $s$th random effect consider $\tilde{\boldsymbol{\gamma}}_s = (\gamma_{s1}, \ldots, \gamma_{sn})^T$ with covariates $\boldsymbol{X}_{cs} \in \mathrm{Mat}_{\mathbb{R}}(n, p_s)$ $\mathrm{Mat}_{\mathbb{R}}(n, m)$ denotes the space of all $n \times m$ matrices with values in $\mathbb{R}$. where $p_s$ denotes the total number of correction-covariates used for the $s$th random effect. Note that $\boldsymbol{X}_{cs}$ has $n$ rows as it contains only one representative observation from each cluster. The correction matrix contains all cluster-constant covariates for random intercepts and just a column of ones for random slopes, which corresponds to centering the given random slope. The correction matrix $\boldsymbol{C}_s$ for the $s$th random effect is obtained by

$$\boldsymbol{C}_s = \boldsymbol{X}_{cs}(\boldsymbol{X}_{cs}^T\boldsymbol{X}_{cs})^{-1}\boldsymbol{X}_{cs}, \quad s = 1, \cdots, q$$

so that the product $(\boldsymbol{I}_n - \boldsymbol{C}_s)\tilde{\boldsymbol{\gamma}}_s$ corrects the $s$th random effect for any covariates contained in the corresponding matrix $\boldsymbol{X}_{cs}$ by counting out the orthogonal projections of the $s$th random effect estimates on the subspace generated by the covariates $\boldsymbol{X}_{cs}$. This ensures the coefficient estimate for the random effects to be uncorrelated with any observed covariate. These separate corrections are summarised in one single correction matrix by defining the block diagonal $\tilde{\boldsymbol{C}} = \mathrm{diag}(\boldsymbol{C}_1, \ldots, \boldsymbol{C}_q)$ and computing

$$\boldsymbol{C} = \boldsymbol{P}^{-1}(\boldsymbol{I}_{nq} - \tilde{\boldsymbol{C}})\boldsymbol{P},$$

where $\boldsymbol{P}$ is a permutation matrix mapping $\gamma$ to

$$\boldsymbol{P}\gamma = \tilde{\gamma} = (\tilde{\gamma}_1^T, \cdots, \tilde{\gamma}_q^T)^T.$$

The product $\boldsymbol{C}_{\gamma}$ then corrects every random effect simultaneously. This concept also proved useful for an improved estimation of mixed models via model-based gradient-boosting [30].

**2.3.5 Updating variance-covariance-components.** The covariance matrix $\boldsymbol{Q}$ of the random effects is updated with an approximate EM-algorithm using the posterior curvatures $\boldsymbol{F}_i$ of the random effects model [31]. An update is received by computing

$$\hat{\boldsymbol{Q}} = \frac{1}{n}\sum_{i=1}^{n}(\boldsymbol{F}_i^{-1} + \hat{\gamma}_i\hat{\gamma}_i^T).$$

Each iteration's model error is obtained by

$$\hat{\sigma}^{2[m]} = \mathrm{Var}(\boldsymbol{y} - \hat{\boldsymbol{\eta}}^{[m]}).$$

**2.3.6 Choice of steplength.** The steplength $0 < v \leq 1$ controls the *weakness* of each update and is substantial in order to avoid overfitting and give each candidate variable equal opportunity to get selected. We stick to the choice of $v = 0.1$ for both fixed and random effects updates, which is well established in the boosting community and thus makes a fairer comparison. This ensures that neither of the coefficient estimates is growing too quickly.

**2.3.7 Stopping iteration.** The algorithm is stopped based on AIC or BIC, i.e.

$$m_* = \operatorname*{arg\,min}_{m=1,\cdots,m_{\text{stop}}} \text{AIC}^{[m]},$$

$$m_* = \operatorname*{arg\,min}_{m=1,\cdots,m_{\text{stop}}} \text{BIC}^{[m]},$$

where $\text{AIC}^{[m]}$ and $\text{BIC}^{[m]}$ denote the information criteria after $m$ iterations. An alternative and computationally more burdensome stopping rule would rely on cluster-wise cross-validation [32], which is however asymptotically equivalent to the marginal AIC as used above [33].

# 3 Simulation study

The algorithm is evaluated with a simulation study. The single simulation scenarios are described in the first subsection, while results are discussed in the latter two. Primary focus is to show, that the algorithm solves the identification problem of the random effects and thus is compared to the bGLMM function of the GMMBoost package available on CRAN. Furthermore, its performance is compared to the classical method implemented in the lmer function of the lme4 package as well as the glmmLasso function of the same-named package, which is another popular approach to regularized regression with potentially high numbers of candidate variables. Please note that we did not include mboost in the comparison, as its approach to random effects is not able to estimate variance components for random intercepts not to mention covariance matrices for multiple slopes. The bGLMM was also compared to the glmmPQL function [4] in [19].

The comparison focuses on mean squared errors of estimates for fixed effects and the random structure as an indicator for overall performance and to address the identification problem. Variable selection properties are evaluated via true and false positive as well as well as false discovery rates. As a side note, we compare computational effort.

## 3.1 Setups

The first setups' random structure consists of random intercepts only. Overall, the setup includes four informative covariates and in addition varying numbers of non-informatives. For $i = 1, \ldots, 50$ and $j = 1, \ldots, 5$ we consider the random intercepts setup

$$y_{ij} = \beta_0 + \beta_1 x_{i1} + \beta_2 x_{i2} + \beta_3 x_{ij3} + \beta_4 x_{ij4} + \sum_{r=5}^{P} \beta_r x_{ijr} + \gamma_{0i} + \varepsilon_{ij} \qquad (4)$$

with values $\beta_0 = 1, \beta_1 = 2, \beta_2 = 4, \beta_3 = 3, \beta_4 = 5$ and $\beta_r = 0, r > 5$ for the fixed effects, $x_{ir}, x_{ijr} \sim \mathcal{N}(0, 1)$ for the cluster-constant and cluster-varying covariates and $\gamma_{0i} \sim \mathcal{N}(0, \tau^2)$ and $\varepsilon_{ij} \sim \mathcal{N}(0, \sigma^2)$ for the random components with $\sigma = 0.4$ and $\tau \in \{0.4, 0.8, 1.6\}$. The total amount of covariates is evaluated for the six different cases $p \in \{10, 25, 50, 100, 250, 500\}$ ranging from low to high dimensional setups.

The second setup is a slightly altered scenario with two additional random slopes, one for a cluster-constant and one for a cluster-varying covariate, i.e.

$$
\begin{aligned}
y_{ij} \quad &= \beta_0 + \beta_1 x_{i1} + \beta_2 x_{i2} + \beta_3 x_{ij3} + \beta_4 x_{ij4} + \sum_{r=5}^{P} \beta_r x_{ijr} \\
&\quad + \gamma_{0i} + \gamma_{1i} x_{i2} + \gamma_{2i} x_{ij4} + \varepsilon_{ij}
\end{aligned}
\tag{5}
$$

with

$$
(\gamma_{0i}, \gamma_{1i}, \gamma_{2i}) \sim \mathcal{N}^{\otimes 3}(0, \boldsymbol{Q}), \quad \boldsymbol{Q} := \begin{pmatrix} \tau^2 & \tau^* & \tau^* \\ \tau^* & \tau^2 & \tau^* \\ \tau^* & \tau^* & \tau^2 \end{pmatrix},
$$

where $\tau \in \{0.4, 0.8, 1.6\}$ and $\tau^*$ is chosen so that $\mathrm{cor}(\gamma_{ki}, \gamma_{li}) = 0.6$ for all $k, l = 1, 2, 3$ holds.

For $\boldsymbol{\beta} = (\beta_0, \ldots, \beta_p)^T$ we consider mean squared errors

$$
\mathrm{mse}_{\boldsymbol{\beta}} := \|\boldsymbol{\beta} - \hat{\boldsymbol{\beta}}\|^2, \quad \mathrm{mse}_{\sigma} := (\sigma - \hat{\sigma})^2, \quad \mathrm{mse}_{\tau} := (\tau - \hat{\tau})^2, \quad \mathrm{mse}_{Q} := \|\boldsymbol{Q} - \hat{\boldsymbol{Q}}\|_F^2
$$

as an indicator for estimation accuracy with $\|\cdot\|_F$ denoting the Frobenius norm of a given matrix. Variable selection properties are evaluated by calculating false positives (FP), true positives (TP) and false discovery rates (FDR)

$$
\mathrm{FP} = \frac{\bar{P}_{\mathrm{sel}}}{\bar{P}_{\mathrm{tot}}}, \quad \mathrm{TP} = \frac{P_{\mathrm{sel}}}{P_{\mathrm{tot}}} \quad \text{and} \quad \mathrm{FDR} = \frac{\bar{P}_{\mathrm{sel}}}{P_{\mathrm{sel}} + \bar{P}_{\mathrm{sel}}},
$$

where $P_{\mathrm{sel}}$ and $P_{\mathrm{tot}}$ denote the amounts of selected informative and total informative candidate variables with $\bar{P}_{\mathrm{sel}}$ and $\bar{P}_{\mathrm{tot}}$ as the equivalents for non-informative covariates. Finally, the elapsed time is measured in seconds where each simulation run was carried out on a *2 x 2.66 GHz-6-Core Intel Xeon* CPU (*64GB* RAM).

Every single simulation setup was independently executed 100 times and, in order to account for skewness of the mean squared error distributions, median values are reported for estimation accuracy and average values for variable selection properties and computation time. The `bGLMM` boosting algorithm was initialized with $m_{\mathrm{stop}} = 500$, while `lbbLMM` was iterated up to $m_{\mathrm{stop}} = 1500$. To determine the optimal penalization parameter for `glmmLasso`, the grid $\{500, 495, 490, \ldots, 0\}$ was used. All of the included regularization approaches were tuned using the BIC.

## 3.2 Results: Random intercepts

**3.2.1 Estimation accuracy.**   Table 1 summarizes results for estimation accuracy. In general, the `lbbLMM` algorithm produces very precise estimates while the `bGLMM` function suffers from the prescribed identification problem yielding a minimum mean squared error of $2^2 + 4^2 = 20$ as the cluster-constant covariates are not being selected. All methods get less precise as the values of $\tau$ and $p$ increase, only `lbbLMM` has stable error rates regarding the amount of candidate variables $p$. Overall, `lbbLMM` outperforms its competitors in every single scenario. Estimation accuracy of the random structure is described in Table 2. Estimates by `lbbLMM` behave similarly well as by `lmer` while the identification problem in `bGLMM` results in high error rates. While lying in the same range as `lmer`, `lbbLMM` clearly outperforms the remaining regularization approaches.

**Table 1. Median mse$_\beta$ of 100 independent simulation runs for each random intercepts setup with corresponding interquartile range.**

| | | lmer | | glmmLasso | | bGLMM | | lbbLMM | |
|---|---|---|---|---|---|---|---|---|---|
| $\tau$ | p | mse$_\beta$ | (iqr) | mse$_\beta$ | (iqr) | mse$_\beta$ | (iqr) | mse$_\beta$ | (iqr) |
| 0.4 | 10 | 0.018 | 0.01 | 0.026 | 0.02 | 20.134 | 0.40 | 0.012 | 0.01 |
| 0.4 | 25 | 0.032 | 0.01 | 0.044 | 0.04 | 20.168 | 0.44 | 0.012 | 0.01 |
| 0.4 | 50 | 0.060 | 0.02 | 0.212 | 0.45 | 20.155 | 0.41 | 0.014 | 0.01 |
| 0.4 | 100 | 0.151 | 0.04 | 1.097 | 0.92 | 20.198 | 0.52 | 0.014 | 0.01 |
| 0.4 | 250 | - | - | 1.917 | 1.61 | 20.191 | 0.55 | 0.011 | 0.01 |
| 0.4 | 500 | - | - | 2.759 | 2.89 | 20.172 | 0.42 | 0.010 | 0.01 |
| 0.8 | 10 | 0.046 | 0.04 | 0.060 | 0.05 | 20.177 | 0.48 | 0.040 | 0.04 |
| 0.8 | 25 | 0.064 | 0.05 | 0.121 | 0.08 | 20.175 | 0.46 | 0.044 | 0.04 |
| 0.8 | 50 | 0.092 | 0.05 | 0.260 | 0.30 | 20.194 | 0.54 | 0.046 | 0.05 |
| 0.8 | 100 | 0.185 | 0.05 | 1.069 | 0.78 | 20.185 | 0.45 | 0.037 | 0.04 |
| 0.8 | 250 | - | - | 2.243 | 1.84 | 20.174 | 0.42 | 0.035 | 0.04 |
| 0.8 | 500 | - | - | 2.459 | 2.96 | 20.114 | 0.30 | 0.035 | 0.03 |
| 1.6 | 10 | 0.149 | 0.17 | 0.175 | 0.19 | 20.194 | 0.51 | 0.145 | 0.17 |
| 1.6 | 25 | 0.190 | 0.18 | 0.215 | 0.20 | 20.190 | 0.57 | 0.170 | 0.18 |
| 1.6 | 50 | 0.187 | 0.17 | 0.265 | 0.30 | 20.174 | 0.48 | 0.137 | 0.17 |
| 1.6 | 100 | 0.282 | 0.17 | 0.819 | 0.93 | 20.214 | 0.49 | 0.147 | 0.17 |
| 1.6 | 250 | - | - | 2.204 | 2.04 | 20.126 | 0.34 | 0.130 | 0.14 |
| 1.6 | 500 | - | - | 3.303 | 2.70 | 20.331 | 0.93 | 0.117 | 0.14 |

**3.2.2 Variable selection.** Table 3 depicts variable selection properties. While selection quality of glmmLasso improves as *p* increases, both boosting approaches yield perfect properties with respect to false positives. However, the identification problem of bGLMM leads to a low true positives rate, as informative effects of cluster-constant covariates are being captured

**Table 2. Median mse$_\tau$ and mse$_\sigma$ of 100 independent simulation runs for each random intercepts setup.**

| | | lmer | | glmmLasso | | bGLMM | | lbbLMM | |
|---|---|---|---|---|---|---|---|---|---|
| $\tau$ | p | mse$_\tau$ | mse$_\sigma$ | mse$_\tau$ | mse$_\sigma$ | mse$_\tau$ | mse$_\sigma$ | mse$_\tau$ | mse$_\sigma$ |
| 0.4 | 10 | 0.001 | 2e-04 | 0.002 | 0.013 | 2.911 | 0.002 | 0.001 | 0.002 |
| 0.4 | 25 | 0.001 | 2e-04 | 0.003 | 0.012 | 2.930 | 0.002 | 0.001 | 0.002 |
| 0.4 | 50 | 0.001 | 2e-04 | 0.006 | 0.051 | 2.970 | 0.002 | 0.001 | 0.002 |
| 0.4 | 100 | 0.002 | 4e-04 | 0.012 | 0.409 | 3.001 | 0.002 | 0.001 | 0.002 |
| 0.4 | 250 | - | - | 0.014 | 0.693 | 2.994 | 0.002 | 0.001 | 0.002 |
| 0.4 | 500 | - | - | 0.014 | 1.108 | 2.948 | 0.002 | 0.001 | 0.002 |
| 0.8 | 10 | 0.003 | 2e-04 | 0.096 | 0.137 | 1.750 | 0.002 | 0.003 | 0.002 |
| 0.8 | 25 | 0.002 | 2e-04 | 0.098 | 0.142 | 1.782 | 0.002 | 0.002 | 0.002 |
| 0.8 | 50 | 0.004 | 2e-04 | 0.090 | 0.165 | 1.797 | 0.002 | 0.005 | 0.002 |
| 0.8 | 100 | 0.004 | 2e-04 | 0.081 | 0.617 | 1.781 | 0.002 | 0.003 | 0.002 |
| 0.8 | 250 | - | - | 0.081 | 1.191 | 1.769 | 0.002 | 0.003 | 0.002 |
| 0.8 | 500 | - | - | 0.078 | 1.292 | 1.706 | 0.002 | 0.004 | 0.002 |
| 1.6 | 10 | 0.008 | 2e-04 | 0.204 | 0.008 | 0.328 | 0.002 | 0.010 | 0.002 |
| 1.6 | 25 | 0.012 | 2e-04 | 0.294 | 0.041 | 0.336 | 0.002 | 0.014 | 0.002 |
| 1.6 | 50 | 0.012 | 2e-04 | 0.481 | 0.156 | 0.334 | 0.002 | 0.014 | 0.002 |
| 1.6 | 100 | 0.015 | 4e-04 | 1.117 | 1.064 | 0.343 | 0.002 | 0.014 | 0.002 |
| 1.6 | 250 | - | - | 1.160 | 2.336 | 0.310 | 0.002 | 0.015 | 0.002 |
| 1.6 | 500 | - | - | 1.161 | 2.777 | 0.311 | 0.002 | 0.022 | 0.002 |

**Table 3. Variable selection properties averaged over 300 runs for each p-dimensional random intercepts setup.** True positives (TP), false positives (FP) and false discovery rate (FDR). Since no noticeable variability regarding the choice of $\tau$ occurred, results are summarized.

| | glmmLasso | | | bGLMM | | | lbbLMM | | |
|---|---|---|---|---|---|---|---|---|---|
| **p** | **FP** | **TP** | **FDR** | **FP** | **TP** | **FDR** | **FP** | **TP** | **FDR** |
| 10 | 0.83 | 1.00 | 0.55 | 0.00 | 0.65 | 0.00 | 0.00 | 1.00 | 0.00 |
| 25 | 0.90 | 1.00 | 0.80 | 0.00 | 0.64 | 0.00 | 0.00 | 1.00 | 0.00 |
| 50 | 0.73 | 1.00 | 0.78 | 0.00 | 0.64 | 0.00 | 0.00 | 1.00 | 0.00 |
| 100 | 0.29 | 1.00 | 0.60 | 0.00 | 0.64 | 0.00 | 0.00 | 1.00 | 0.00 |
| 250 | 0.05 | 1.00 | 0.57 | 0.00 | 0.65 | 0.00 | 0.00 | 1.00 | 0.00 |
| 500 | 0.02 | 1.00 | 0.61 | 0.00 | 0.65 | 0.00 | 0.00 | 1.00 | 0.00 |

by the random intercepts. lbbLMM on the other hand has perfect selection properties with respect to both, true and false positives.

## 3.3 Results: Random slopes

**3.3.1 Estimation accuracy.** Table 4 summarizes results for estimation accuracy. Except for the increased error rates in general, the behaviour is similar to the random intercepts setup (4). lbbLMM again performs stable as $p$ increases and clearly outperforms the other regularization approaches. However, lmer has slightly better error rates in low-dimensional scenarios with higher values for $\tau$, which can be also seen in Table 5.

**3.3.2 Variable selection.** Table 6 depicts variable selection properties for the random slopes setup. Results are almost identical to the random intercepts setup described in Table 3.

The presence of the identification problem in bGLMM is reflected in high errors for fixed and random effects and low true positives rates. Based on its good values for $mse_\beta$, $mse_\tau$, $mse_Q$ and TP it can be stated that lbbLMM not only solves the problems occurring in bGLMM but

**Table 4. Median $mse_\beta$ of 100 independent simulation runs for each random slopes setup with corresponding interquartile range.**

| | | lmer | | glmmLasso | | bGLMM | | lbbLMM | |
|---|---|---|---|---|---|---|---|---|---|
| **$\tau$** | **p** | **$mse_\beta$** | **(iqr)** | **$mse_\beta$** | **(iqr)** | **$mse_\beta$** | **(iqr)** | **$mse_\beta$** | **(iqr)** |
| 0.4 | 10 | 0.025 | 0.02 | 0.064 | 0.06 | 40.955 | 1.98 | 0.023 | 0.03 |
| 0.4 | 25 | 0.043 | 0.02 | 0.096 | 0.08 | 40.979 | 1.97 | 0.026 | 0.03 |
| 0.4 | 50 | 0.083 | 0.03 | 0.233 | 0.31 | 41.017 | 1.94 | 0.027 | 0.04 |
| 0.4 | 100 | 0.204 | 0.06 | 1.395 | 0.94 | 41.067 | 1.94 | 0.028 | 0.04 |
| 0.4 | 250 | - | - | 2.552 | 1.72 | 41.077 | 1.77 | 0.031 | 0.04 |
| 0.4 | 500 | - | - | 2.578 | 2.43 | 40.807 | 1.38 | 0.022 | 0.03 |
| 0.8 | 10 | 0.068 | 0.06 | 0.125 | 0.12 | 41.048 | 1.86 | 0.084 | 0.12 |
| 0.8 | 25 | 0.091 | 0.06 | 0.176 | 0.18 | 41.048 | 1.80 | 0.088 | 0.14 |
| 0.8 | 50 | 0.133 | 0.07 | 0.241 | 0.27 | 41.185 | 1.64 | 0.101 | 0.16 |
| 0.8 | 100 | 0.300 | 0.10 | 1.040 | 1.14 | 41.113 | 1.53 | 0.107 | 0.14 |
| 0.8 | 250 | - | - | 2.323 | 2.01 | 40.985 | 1.59 | 0.082 | 0.10 |
| 0.8 | 500 | - | - | 3.320 | 2.46 | 40.917 | 1.59 | 0.099 | 0.12 |
| 1.6 | 10 | 0.260 | 0.22 | 0.473 | 0.53 | 41.741 | 3.17 | 0.346 | 0.49 |
| 1.6 | 25 | 0.297 | 0.22 | 0.457 | 0.52 | 41.751 | 2.96 | 0.413 | 0.59 |
| 1.6 | 50 | 0.311 | 0.26 | 0.532 | 0.51 | 41.340 | 2.92 | 0.446 | 0.53 |
| 1.6 | 100 | 0.547 | 0.33 | 0.903 | 0.74 | 41.211 | 2.32 | 0.351 | 0.45 |
| 1.6 | 250 | - | - | 3.860 | 3.31 | 41.794 | 2.79 | 0.410 | 0.48 |
| 1.6 | 500 | - | - | 4.364 | 4.10 | 41.711 | 3.17 | 0.337 | 0.26 |

**Table 5. Median $mse_Q$ and $mse_\sigma$ of 100 independent simulation runs for each random slopes setup.**

| | | lmer | | glmmLasso | | bGLMM | | lbbLMM | |
|---|---|---|---|---|---|---|---|---|---|
| $\tau$ | p | $mse_Q$ | $mse_\sigma$ | $mse_Q$ | $mse_\sigma$ | $mse_Q$ | $mse_\sigma$ | $mse_Q$ | $mse_\sigma$ |
| 0.4 | 10 | 0.017 | 3e-04 | 0.092 | 0.029 | 888.217 | 0.008 | 0.017 | 0.004 |
| 0.4 | 25 | 0.017 | 4e-04 | 0.091 | 0.032 | 894.852 | 0.009 | 0.017 | 0.004 |
| 0.4 | 50 | 0.017 | 4e-04 | 0.082 | 0.057 | 901.488 | 0.009 | 0.016 | 0.004 |
| 0.4 | 100 | 0.025 | 8e-04 | 0.074 | 0.469 | 902.736 | 0.009 | 0.017 | 0.004 |
| 0.4 | 250 | - | - | 0.074 | 1.039 | 872.132 | 0.009 | 0.019 | 0.004 |
| 0.4 | 500 | - | - | 0.076 | 1.073 | 830.769 | 0.008 | 0.016 | 0.003 |
| 0.8 | 10 | 0.196 | 3e-04 | 1.543 | 0.092 | 857.413 | 0.009 | 0.245 | 0.002 |
| 0.8 | 25 | 0.220 | 4e-04 | 1.538 | 0.136 | 885.362 | 0.009 | 0.237 | 0.002 |
| 0.8 | 50 | 0.222 | 2e-04 | 1.583 | 0.223 | 883.447 | 0.009 | 0.256 | 0.002 |
| 0.8 | 100 | 0.245 | 7e-04 | 1.744 | 0.627 | 892.735 | 0.009 | 0.237 | 0.002 |
| 0.8 | 250 | - | - | 1.762 | 1.704 | 863.080 | 0.008 | 0.212 | 0.002 |
| 0.8 | 500 | - | - | 1.776 | 2.184 | 901.223 | 0.009 | 0.180 | 0.003 |
| 1.6 | 10 | 3.297 | 3e-04 | 21.546 | 0.052 | 901.261 | 0.008 | 3.823 | 0.003 |
| 1.6 | 25 | 3.207 | 4e-04 | 21.314 | 0.186 | 911.947 | 0.009 | 3.823 | 0.004 |
| 1.6 | 50 | 3.112 | 4e-04 | 23.035 | 0.331 | 889.823 | 0.009 | 3.761 | 0.005 |
| 1.6 | 100 | 3.338 | 5e-04 | 27.822 | 1.024 | 875.406 | 0.008 | 3.043 | 0.004 |
| 1.6 | 250 | - | - | 29.964 | 2.889 | 940.885 | 0.009 | 2.779 | 0.004 |
| 1.6 | 500 | - | - | 32.257 | 5.413 | 844.134 | 0.008 | 3.026 | 0.004 |

also offers a reliable and good performing regularization approach to linear mixed models in general.

## 3.4 Computation time

Table 7 depicts average computation times for the random intercepts (4) and random slopes (5) setup. All regularization approaches roughly show a linear scaling with increasing amount of candidate variables *p*. In most cases, glmmLasso runs noticeably faster than its two boosting competitors. However, a direct comparison is hard to interpret as the computation time of glmmLasso strongly depends on the fineness of the grid used in order to determine the optimal penalization parameter. In addition, the corrupt updating process of bGLMM leads to substantial faster convergence, as the algorithm is due to its identification issue capable of fitting multiple effects in one single iteration and thus achieves faster convergence, which is also the reason why bGLMM runs faster in the random slopes setup.

**Table 6. Variable selection properties averaged over 300 runs for each p-dimensional random slopes setup.** True positives (TP), false positives (FP) and false discovery rate (FDR). Since no noticeable variability regarding the choice of $\tau$ occurred, results are summarized.

| | glmmLasso | | | bGLMM | | | lbbLMM | | |
|---|---|---|---|---|---|---|---|---|---|
| p | FP | TP | FDR | FP | TP | FDR | FP | TP | FDR |
| 10 | 0.81 | 1.00 | 0.55 | 0.00 | 0.68 | 0.00 | 0.00 | 1.00 | 0.00 |
| 25 | 0.94 | 1.00 | 0.83 | 0.00 | 0.68 | 0.00 | 0.00 | 1.00 | 0.00 |
| 50 | 0.90 | 1.00 | 0.89 | 0.00 | 0.68 | 0.00 | 0.00 | 1.00 | 0.00 |
| 100 | 0.61 | 1.00 | 0.80 | 0.00 | 0.67 | 0.00 | 0.00 | 1.00 | 0.00 |
| 250 | 0.21 | 1.00 | 0.68 | 0.00 | 0.67 | 0.00 | 0.00 | 1.00 | 0.00 |
| 500 | 0.03 | 1.00 | 0.66 | 0.00 | 0.67 | 0.00 | 0.00 | 1.00 | 0.00 |

**Table 7. Averages of elapsed computation time of 100 independent simulation runs for each random intercepts ($t_{int}$) and slopes ($t_{slp}$) setup.**

| | | lmer | | glmmLasso | | bGLMM | | lbbLMM | |
|---|---|---|---|---|---|---|---|---|---|
| $\tau$ | p | $t_{int}$ | $t_{slp}$ | $t_{int}$ | $t_{slp}$ | $t_{int}$ | $t_{slp}$ | $t_{int}$ | $t_{slp}$ |
| 0.4 | 10 | 0.14 | 0.26 | 16.24 | 54.16 | 61.34 | 52.75 | 58.53 | 137.17 |
| 0.4 | 25 | 0.15 | 0.35 | 32.19 | 83.51 | 142.45 | 119.60 | 96.17 | 217.57 |
| 0.4 | 50 | 0.18 | 0.51 | 58.11 | 131.92 | 281.60 | 229.62 | 159.68 | 363.74 |
| 0.4 | 100 | 0.27 | 1.19 | 121.26 | 233.59 | 551.97 | 449.22 | 276.32 | 674.16 |
| 0.4 | 250 | - | - | 304.95 | 451.52 | 1352.46 | 1103.51 | 663.02 | 1735.15 |
| 0.4 | 500 | - | - | 585.84 | 863.54 | 2739.12 | 2195.73 | 1344.99 | 2799.55 |
| 0.8 | 10 | 0.13 | 0.28 | 20.21 | 93.14 | 61.00 | 52.72 | 59.88 | 123.91 |
| 0.8 | 25 | 0.15 | 0.36 | 35.33 | 108.02 | 142.30 | 119.11 | 97.24 | 195.48 |
| 0.8 | 50 | 0.18 | 0.57 | 59.34 | 154.90 | 276.57 | 229.47 | 155.17 | 338.10 |
| 0.8 | 100 | 0.27 | 1.27 | 114.51 | 239.32 | 539.65 | 449.19 | 281.19 | 608.19 |
| 0.8 | 250 | - | - | 296.17 | 434.89 | 1379.50 | 1099.60 | 685.09 | 1246.12 |
| 0.8 | 500 | - | - | 586.13 | 863.36 | 2706.37 | 2194.56 | 1323.87 | 2541.26 |
| 1.6 | 10 | 0.14 | 0.37 | 45.36 | 207.99 | 60.11 | 52.42 | 60.25 | 130.47 |
| 1.6 | 25 | 0.17 | 0.50 | 51.06 | 212.15 | 140.23 | 118.34 | 95.72 | 195.48 |
| 1.6 | 50 | 0.19 | 0.79 | 72.13 | 236.06 | 269.85 | 229.78 | 158.76 | 321.88 |
| 1.6 | 100 | 0.29 | 1.73 | 124.38 | 284.19 | 542.70 | 451.25 | 291.27 | 587.04 |
| 1.6 | 250 | - | - | 278.47 | 429.77 | 1368.52 | 1098.15 | 682.92 | 1272.76 |
| 1.6 | 500 | - | - | 583.67 | 873.84 | 2725.24 | 2194.31 | 1396.19 | 2623.04 |

However, the computational effort for bGLMM is very sensitive to increasing numbers of total observations $N$. Table 8 and Fig 2 depict averaged computation times of the random intercepts scenario (4) with $\tau = 0.4$ and $p = 10$ fixed, but varying values $n_i \in \{5, 10, 15, 20\}$, i.e. the number of observations per cluster. While glmmLasso and lbbLMM have a linear relationship between $N$ and elapsed computation time, bGLMM increases exponentially making the method less applicable even to data sets with fewer candidate variables when the number of total observations is large.

## 4 Primary biliary cirrhosis

The primary biliary cirrhosis (PBC) dataset from 1994 [34] tracks the change of the serum bilirubin level for a total of 312 PBC patients randomized into a treatment and a placebo group and additionally contains baseline covariates as well as follow-up measurements of several biomarkers. The dataset is, among others, available in the JM package [35] and Table 9 gives an overview of the single covariates included in the data and how they are coded in the model formula. The serum bilirubin level, here modelled as the response variable, is considered a strong indicator for disease progression, hence an appropriate quantification of the impact of the

**Table 8. Averages of elapsed computation time of 100 independent simulation runs regarding varying values for $n_i$ with $\tau = 0.4$ and $p = 10$ fixed.**

| | | lmer | glmmLasso | bGLMM | lbbLMM |
|---|---|---|---|---|---|
| $n_i$ | N | $t_{int}$ | $t_{int}$ | $t_{int}$ | $t_{int}$ |
| 5 | 250 | 0.13 | 15.21 | 60.14 | 59.04 |
| 10 | 250 | 0.14 | 32.51 | 978.16 | 72.25 |
| 15 | 750 | 0.16 | 56.92 | 5799.48 | 113.55 |
| 20 | 1000 | 0.16 | 84.46 | 13841.07 | 172.29 |

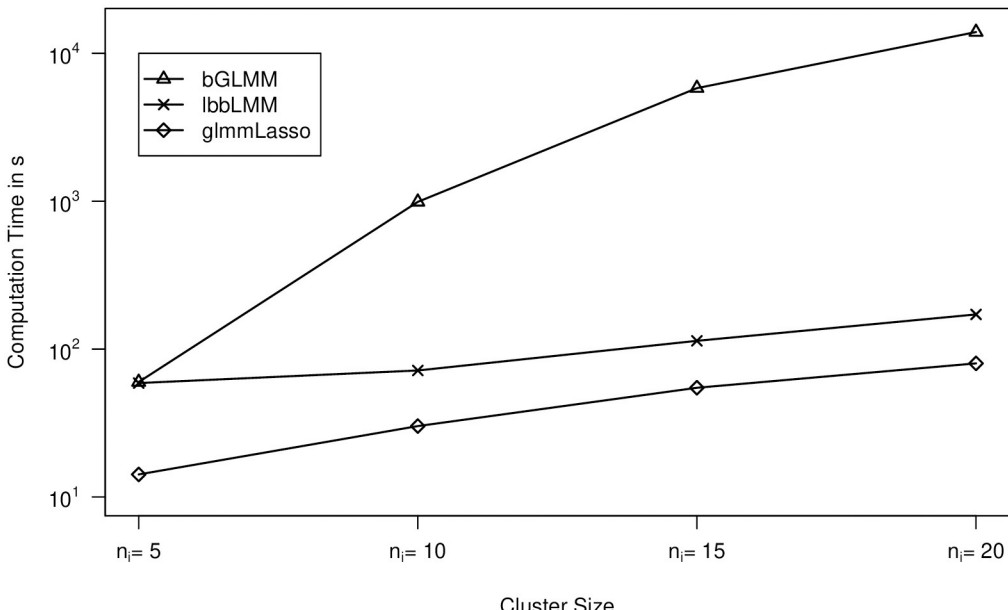

**Fig 2. Differing computational effort of the regularization routines `bGLMM`, `lbbLMM` and `glmmLasso` for varying cluster sizes.**

given covariates on the serum bilirubin level will lead to an adequate prediction model for the health status of PBC patients. Using boosting to carry out this quantification will optimize the prediction properties. For $y_{ij}$ denoting the $j$th measurement of serum bilirubin for the $i$th patient, we formulate the random intercept model

$$
\begin{aligned}
y_{ij} &= \beta_0 + \beta_1 \texttt{drug}_i + \beta_2 \texttt{age}_i + \beta_3 \texttt{sex}_i + \beta_4 \texttt{asc}_i \\
&+ \beta_5 \texttt{hep}_i + \beta_6 \texttt{spi}_i + \beta_7 t_{ij} + \beta_8 t_{ij}^2 + \beta_9 \texttt{alb}_{ij} \\
&+ \beta_{10} \texttt{alk}_{ij} + \beta_{11} \texttt{SGOT}_{ij} + \beta_{12} \texttt{pla}_{ij} + \beta_{13} \texttt{pro}_{ij} + \gamma_{0i} + \varepsilon_{ij}
\end{aligned}
\tag{6}
$$

**Table 9. Variables of the PBC data set.** `drug` and `sex` are dummies for treatment group and female gender. Ascites is the abnormal buildup of fluid in the abdomen and spiders are blood vessel malformations in the skin. SGOT is short for serum glutamic oxaloacetic transaminase.

| time-constant | continuous | age at baseline | age |
|---|---|---|---|
| | discrete | treatment group gender | drug |
| | | gender | sex |
| time-varying | continuous | albumin | alb |
| | | alkaline | alk |
| | | SGOT | SGOT |
| | | platelet count | pla |
| | | prothrombin time | pro |
| | | time in years | t |
| | discrete | ascites | asc |
| | | spiders | spi |
| | | enlarged liver | hep |

**Table 10. Variable selection and shrinkage of various regularization approaches compared to `lmer`.**

| | glmmLasso | bGLMM | lbbLMM | lmer | P (lmer) |
|---|---|---|---|---|---|
| (Intercept) | 3.94 | 4.16 | 4.05 | 3.95 | < 0.001 |
| drug | -0.23 | - | -0.24 | -0.24 | 0.26 |
| age | -0.12 | - | - | -0.15 | 0.49 |
| sex | 0.07 | - | - | 0.10 | 0.65 |
| asc | 0.79 | 0.69 | 0.71 | 0.79 | < 0.001 |
| hep | 0.35 | 0.23 | 0.26 | 0.35 | < 0.001 |
| spi | 0.41 | 0.30 | 0.30 | 0.39 | < 0.001 |
| $t$ | 0.78 | 1.04 | 0.60 | 0.90 | < 0.001 |
| $t^2$ | -0.30 | -0.37 | - | -0.42 | 0.02 |
| alb | -0.44 | -0.36 | -0.35 | -0.41 | < 0.001 |
| alk | 0.04 | -0.04 | - | 0.08 | 0.35 |
| SGOT | 1.02 | 0.84 | 0.95 | 1.05 | < 0.001 |
| pla | -0.11 | -0.05 | - | -0.10 | 0.33 |
| pro | 0.16 | - | - | 0.19 | 0.01 |
| $\hat{\tau}$ | 3.55 | 4.20 | 3.79 | 3.55 | |
| time | 1465 | 60543 | 101 | 0.15 | |

with $\gamma_{0i} \sim \mathcal{N}(0, \tau^2)$ and an included square time effect, since the effect of time might be non-linear. Both boosting approaches were initialized with $m_{\text{stop}} = 500$ and the grid {0, 0.1, 0.2, . . ., 30} was chosen for `glmmLasso`. Based on BIC, `bGLMM` determined $m^* = 389$ and `lbbLMM` $m^* = 161$ as the best performing number of iterations and the optimal tuning parameter for `glmmLasso` was $\lambda^* = 12.4$. The coefficient estimates are compared to an unregularized model (`lmer`) displayed with corresponding p-values in Table 10 and the well known coefficient paths for `lbbLMM` are depicted in Fig 3.

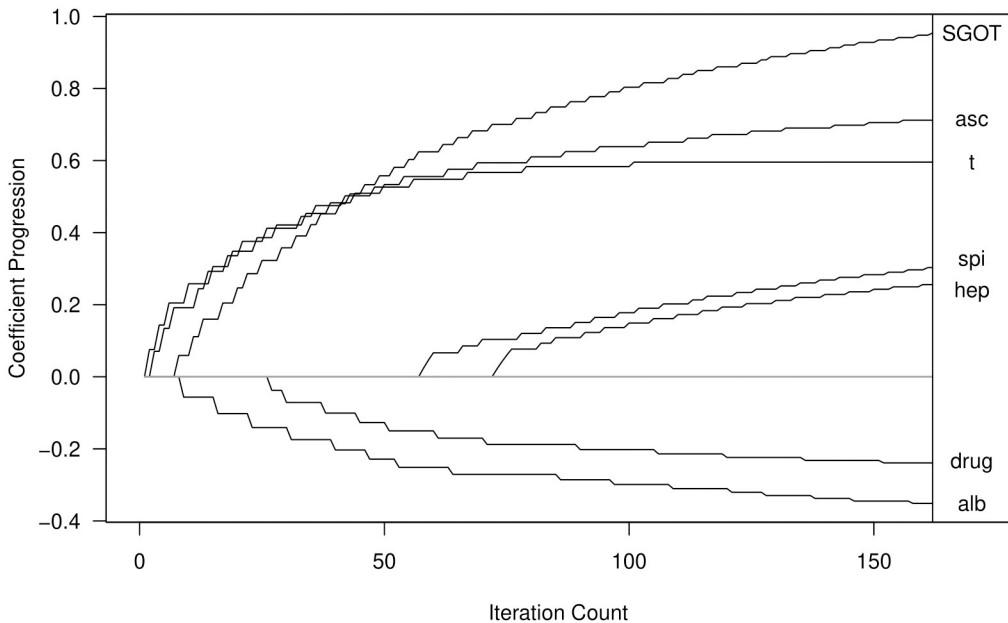

**Fig 3. Coefficient progression for the PBC data obtained by `lbbLMM` with $m_* = 161$.**

In general, the results reflect what was already observed in the simulation study. `glmmLasso` struggles with proper variable selection in lower dimensional scenarios and bGLMM does not select any cluster-constant covariates due to misspecification. Although the effect of `drug` has a comparatively high p-value, the coefficient estimate by bGLMM stands out among all regularization approaches while the value for $\tau$ is simultaneously pretty large, which indicates possible bias arising from wrongly identified random intercepts. On the other hand, the rest of the variables which were selected by `lbbLMM` tend to be of high impact depending on the chosen significance level. For lower choices, e.g. $\alpha \in \{0.01, 0.005\}$ [36], the selection process of `lbbLMM` matches with the selected covariate having significant impact while at the same time receiving shrinkage by the regularization mechanism. Regarding computational effort, bGLMM runs with approximately 17 hours (60543 seconds) tremendously long and needs around 600 times more computation time than its direct competitor `lbbLMM`.

## 5 Discussion

The updated algorithm is due to its minor and major tweaks capable of dealing with cluster-constant covariates in linear mixed models by preventing the random effects from taking up too much space. In addition, it preserves the well-known advantages of boosting techniques in general by offering variable selection and a good functionality even in high dimensional setups. As a very important side effect the computational effort receives a tremendous decrease making the algorithm more applicable to real world scenarios.

Primary hindrance of the `lbbLMM` algorithm is a missing approach for model choice as the random effects structure has to be specified in advance and does not underlie any selection process. Although reasonable options regarding the random structure are limited in most real world applications and could also be evaluated afterwards using appropriate information criteria, it remains and interesting question, how one could incorporate proper model selection during the updating process while simultaneously preserving the advantages gained by the `lbbLMM` algorithm.

Canonical extensions of the successful concept include incorporating non-linear predictor functions, i.e. estimation of smooth effects based on P-splines or extending the algorithm from linear mixed models to generalized mixed models to allow more flexible inference for a wider class of data structures. Both have been incorporated in [37] for classical likelihood-based boosting and it is assumed that the proposed tweaks in the present work would improve performance of the more flexible approaches as well.

## Supporting information

**S1 File.**
(ZIP)

## Acknowledgments

Colin Griesbach performed the present work in partial fulfilment of the requirements for obtaining the degree 'Dr. rer. biol. hum.' at the Friedrich-Alexander-Universität Erlangen-Nürnberg.

## Author Contributions

**Conceptualization:** Colin Griesbach.

**Formal analysis:** Colin Griesbach.

**Funding acquisition:** Elisabeth Bergherr.

**Investigation:** Colin Griesbach.

**Methodology:** Colin Griesbach, Andreas Groll, Elisabeth Bergherr.

**Supervision:** Elisabeth Bergherr.

**Visualization:** Colin Griesbach.

**Writing – original draft:** Colin Griesbach.

**Writing – review & editing:** Andreas Groll, Elisabeth Bergherr.

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
