## [Decision Letter · Decision Letter 0]

6 May 2021

PONE-D-21-03209

Addressing cluster-constant covariates in mixed effects models via likelihood-based boosting techniques

PLOS ONE

Dear Dr. Griesbach,

Thank you for submitting your manuscript to PLOS ONE. After careful consideration, we feel that it has merit but does not fully meet PLOS ONE’s publication criteria as it currently stands. Therefore, we invite you to submit a revised version of the manuscript that addresses the points raised during the review process.

If applicable, we recommend that you deposit your laboratory protocols in protocols.io to enhance the reproducibility of your results. Protocols.io assigns your protocol its own identifier (DOI) so that it can be cited independently in the future. For instructions see: http://journals.plos.org/plosone/s/submission-guidelines#loc-laboratory-protocols . Additionally, PLOS ONE offers an option for publishing peer-reviewed Lab Protocol articles, which describe protocols hosted on protocols.io. Read more information on sharing protocols at https://plos.org/protocols?utm_medium=editorial-email&utm_source=authorletters&utm_campaign=protocols .

We look forward to receiving your revised manuscript.

Kind regards,

Feng Chen

Academic Editor

PLOS ONE

Journal Requirements:

3)   Thank you for stating the following in the Competing Interests section:

[The authors have declared that no competing interests exist.].

We note that you received funding from a commercial source: Volkswagen

Reviewers' comments:

Reviewer's Responses to Questions

**Comments to the Author**

1. Is the manuscript technically sound, and do the data support the conclusions?

Reviewer #1: Yes

Reviewer #2: Yes

2. Has the statistical analysis been performed appropriately and rigorously? 

Reviewer #1: Yes

Reviewer #2: Yes

3. Have the authors made all data underlying the findings in their manuscript fully available?

Reviewer #1: Yes

Reviewer #2: No

4. Is the manuscript presented in an intelligible fashion and written in standard English?

Reviewer #1: Yes

Reviewer #2: Yes

5. Review Comments to the Author

Reviewer #1: The topic is interesting and the paper is overall well written. There is no suggestion to this paper, i.e. the authors should consider to make proper graphs out of major results instead of intense texts with tables, this paper would become much more readable to the potential authors.

Reviewer #2: This study proposed a novel and refined boosting algorithm for mixed models, particularly in the presence of cluster-constant covariates. The advantages of the method in terms of estimation accuracy and computational time were well demonstrated by an extensive simulation study. The application potential of the method was also further illustrated by an empirical study using the primary biliary cirrhosis data. Overall, the topic is interesting and worthy of investigation. The conclusions are robust and reliable. Since the method is technically sound and the contributions are substantial, I have no further comments and thus suggest it for publication.

6. PLOS authors have the option to publish the peer review history of their article (what does this mean?). If published, this will include your full peer review and any attached files.

Reviewer #1: No

Reviewer #2: No

---

## [Author Response · Author response to Decision Letter 0]

10 May 2021

The authors would like to thank the two reviewers for the quick and positive feedback. Please find a detailed point-to-point response listed below.

Reviewer 1:

- The topic is interesting and the paper is overall well written. There is no suggestion to this paper, i.e. the authors should consider to make proper graphs out of major results instead of intense texts with tables, this paper would become much more readable to the potential authors.

Response from the authors: Thanks for this comment. We agree that pure tables can be tiring to read and added two additional figures. One showcasing the computational effort and one depicting the coefficient progression for the data example which highlights the boosting mechanism in more detail.

Reviewer 2:

- This study proposed a novel and refined boosting algorithm for mixed models, particularly in the presence of cluster-constant covariates. The advantages of the method in terms of estimation accuracy and computational time were well demonstrated by an extensive simulation study. The application potential of the method was also further illustrated by an empirical study using the primary biliary cirrhosis data. Overall, the topic is interesting and worthy of investigation. The conclusions are robust and reliable. Since the method is technically sound and the contributions are substantial, I have no further comments and thus suggest it for publication.

Response from the authors: Thank you for the positive feedback.

---

## [Decision Letter · Decision Letter 1]

22 Jun 2021

Addressing cluster-constant covariates in mixed effects models via likelihood-based boosting techniques

PONE-D-21-03209R1

Dear Dr. Griesbach,

We’re pleased to inform you that your manuscript has been judged scientifically suitable for publication and will be formally accepted for publication once it meets all outstanding technical requirements.

Kind regards,

Feng Chen

Academic Editor

PLOS ONE

Additional Editor Comments (optional):

Reviewers' comments:

Reviewer's Responses to Questions

**Comments to the Author**

1. If the authors have adequately addressed your comments raised in a previous round of review and you feel that this manuscript is now acceptable for publication, you may indicate that here to bypass the “Comments to the Author” section, enter your conflict of interest statement in the “Confidential to Editor” section, and submit your "Accept" recommendation.

Reviewer #2: All comments have been addressed

2. Is the manuscript technically sound, and do the data support the conclusions?

Reviewer #2: Yes

3. Has the statistical analysis been performed appropriately and rigorously? 

Reviewer #2: Yes

4. Have the authors made all data underlying the findings in their manuscript fully available?

Reviewer #2: No

5. Is the manuscript presented in an intelligible fashion and written in standard English?

Reviewer #2: Yes

6. Review Comments to the Author

Reviewer #2: Since the contributions are substantial, I have no further comments and thus suggest it for publication.

7. PLOS authors have the option to publish the peer review history of their article (what does this mean?). If published, this will include your full peer review and any attached files.

Reviewer #2: No

---

## [Editor Report · Acceptance letter]

29 Jun 2021

PONE-D-21-03209R1

Addressing cluster-constant covariates in mixed effects models via likelihood-based boosting techniques

Dear Dr. Griesbach:

I'm pleased to inform you that your manuscript has been deemed suitable for publication in PLOS ONE. Congratulations! Your manuscript is now with our production department.

Kind regards,

on behalf of

Dr. Feng Chen

Academic Editor

PLOS ONE